# TEA-GCN: Transformer-Enhanced Adaptive Graph Convolutional Network for Traffic Flow Forecasting

**DOI:** 10.3390/s24217086

**Published:** 2024-11-04

**Authors:** Xiaxia He, Wenhui Zhang, Xiaoyu Li, Xiaodan Zhang

**Affiliations:** 1School of Information Science and Technology, Beijing University of Technology, Beijing 100124, China; hexiaxia@emails.bjut.edu.cn (X.H.); zhangxiaodan@bjut.edu.cn (X.Z.); 2School of Information Engineering, Jiangxi Vocational College of Industry & Enginneering, Nanchang 330013, China; aloxi@163.com; 3Aerospace Information Research Institute, Chinese Academy of Sciences, Beijing 100083, China

**Keywords:** graph convolutional networks, traffic flow forecasting, adaptive graph learning

## Abstract

Traffic flow forecasting is crucial for improving urban traffic management and reducing resource consumption. Accurate traffic conditions prediction requires capturing the complex spatial-temporal dependencies inherent in traffic data. Traditional spatial-temporal graph modeling methods often rely on fixed road network structures, failing to account for the dynamic spatial correlations that vary over time. To address this, we propose a Transformer-Enhanced Adaptive Graph Convolutional Network (TEA-GCN) that alternately learns temporal and spatial correlations in traffic data layer-by-layer. Specifically, we design an adaptive graph convolutional module to dynamically capture implicit road dependencies at different time levels and a local-global temporal attention module to simultaneously capture long-term and short-term temporal dependencies. Experimental results on two public traffic datasets demonstrate the effectiveness of the proposed model compared to other state-of-the-art traffic flow prediction methods.

## 1. Introduction

With the rapid development of urban intelligent transportation, the traffic data collection sensors have been widely deployed and applied, and a large amount of traffic data has been accumulated in real time. In addition, accurate traffic flow forecasting is conducive to improving the management of urban traffic systems and reducing the resource consumption. Therefore, how to efficiently use these collected data for traffic flow forecasting is a core issue and research hotpot in the intelligent transportation field. So far, a large number of researchers have carried out extensive researches on the topic of traffic flow forecasting, and have achieved abundant research results.

Early traffic flow forecasting methods focus on analyzing the temporal dependence of data [1,2,3,4], that is, learning the trend of traffic flow from historical observation sequences. For example, Autoregressive Integrated Moving Average model (ARIMA) [5] is a traditional time series modeling method, which is widely used in traffic flow forecasting tasks. The Historical Average model (HA) [6] is the favored model in the industry due to its excellent efficiency and accuracy. Although these temporal-based analysis methods have achieved satisfactory results, they only consider the temporal correlation of sequence data. With the development of urbanization, the traffic data exhibits complex structural characteristics, so the previous temporal correlation modeling methods are ineffective to fully learn the characteristics of traffic network.

Fortunately, graph structures can represent such complex road network structures. Benefiting from the powerful structure capture ability of Graph Neural Network (GNNs) [7,8,9,10], a series of GNN-based traffic flow forecasting methods have been proposed [11,12]. They generally integrate graph neural networks into Recurrent Neural Networks (RNN) or Convolutional Neural Networks (CNN) to capture the complex spatial-temporal dependencies of traffic data [13,14]. For example, ASTGCN [15] simultaneously utilizes GCN to capture spatial features and Temporal Convolutional Network (TCN) to model temporal dependencies. Although these spatial-temporal correlation modeling methods have achieved relatively satisfactory forecasting performance, there still exist two obvious challenges.

First, the traffic data show the characteristics of dynamic, complexity and uncertainty, in which the spatial dependence between roads changes dynamically over time. Existing GNN-based methods ignore other complex relationships in traffic data except for physical connections, which cannot accurately model the spatial-temporal correlation of traffic data. Specifically, explicit physical structures are difficult to accurately reflect the real dependencies and connection strengths between roads, i.e., the missing edge connections between roads with similar traffic conditions. Based on this challenge, Wu et al. [16] constructed an adaptive adjacency matrix and learned it through the node embedding to capture the hidden spatial dependence of traffic data. But they ignored the dynamic of road spatial dependence over time. Learning a fixed spatial relationship from traffic data is not sufficient to reflect the changes in road dependencies. Therefore, how to accurately model the dynamic spatial dependence of traffic data is a challenge.

Second, existing temporal dependency modeling methods are ineffective to deal with the long-term sequence information. The RNN-based methods require multiple iterations to process long-term sequences, resulting in the high computational cost and the problem of gradient disappearance. The CNN-based methods exploit one-dimensional convolution to mine the temporal dependency between sequences, and its receptive field increases linearly with the depth of convolutional layers, which is difficult to capture the long-term correlation. So, how to process long-term traffic data is also a challenge.

To tackle above challenges, in this paper, we propose a novel transformer-enhanced adaptive graph convolutional network for traffic flow forecasting. Its brief illustration is show in Figure 1. Specifically, the adaptive graph convolutional module is proposed to model dynamic spatial dependencies of complex traffic data, and the local-global temporal attention module is proposed to explore the temporal dependencies of traffic data under different periods, which improves the robustness of the proposed model. The contributions of this paper are listed as follows:We propose a novel GNN-based traffic flow forecasting model that alternatively updates the graph structure and the data representation layer-by-layer, which simultaneously captures the complex spatial-temporal dependencies of traffic data.We creatively design an adaptive graph convolutional module to capture the rich and implicit dynamic road spatial dependence under different time levels, which is obviously different from other GNN-based methods.We propose a local-global temporal attention module to model both long-term and short-term temporal dependence of the traffic data.

The rest of this paper mainly consists of the following parts: In Section 2, we review existing traffic forecasting methods. In Section 3, we provide a problem statement. The proposed Transformer-Enhanced Adaptive Graph Convolutional Network is described in detail in Section 4. Then, we verify the effectiveness of the proposed method through a series of experiments in Section 5. Finally, we conclude this paper in Section 6.

## 2. Related Work

In this section, we review several related researches about traffic flow forecasting.

Early traffic flow forecasting methods are mainly based on statistical models. ARIMA [5] is an important method specially designed for time series forecasting. Besides, Kalman filter [1] is also widely used in traffic flow forecasting tasks.

With the popularity of deep learning, many researchers have introduced deep learning into traditional traffic flow forecasting algorithms and achieved the satisfactory performance [17]. Lv et al. [18] proposed a novel traffic flow forecasting method on the basis of deep learning technology, which firstly exploits autoencoder to learn the traffic flow features. To describe the stochastic and nonlinear natures of traffic data, Hu et al. [4] utilized RNN to model the temporal dependence of data, which further verifies the effectiveness of deep learning in traffic flow prediction tasks.

Due to the complex, non-Euclidean structure of traffic data, traditional temporal dependence modeling methods are able to fully extract these characteristics. Rencent research has applied GNNs to traffic flow forecasting tasks, capturing spatial-temporal dependencies and achieving satisfactory performance. One type of research integrating GNNs with RNN or CNN to recursively mine the complex temporal and spatial dependence hidden in traffic data [13,19,20]. Zhao et al. [21] proposed a temporal graph convolutional network for urban road network traffic forecasting, which utilizes GCN to learn spatial dependencies and GRU to learn the trend of traffic data in the time dimension. Graph WaveNet constructs an adaptive adjacency matrix to capture the implicit spatial dependence in traffic data [16]. On this basis, Bai et al. [19] proposed an adaptive graph convolutional recurrent network to capture fine-grained spatial and temporal dependencies in traffic sequences through the adaptive node parameter learning and adaptive graph generation.

Another type of research focuses on developing a spatial-temporal fusion graph for traffic data and using GNNs to capture spatial-temporal correlations synchronously [14,22]. Ref. [23] employs dynamic hypergraph structure learning and interactive graph convolution to capture high-order spatio-temporal relationships and diverse transitional patterns in traffic data. Ref. [24] proposes a novel spatio-temporal graph neural network model that conjointly captures high-order spatio-temporal relationships and diverse traffic patterns within traffic data. To further consider the importance of pivotal nodes, which exhibit more complex spatio-temporal dependencies than other nodes. Ref. [25] proposes a novel pivotal node identification module that identifies and models pivotal nodes and their complex spatio-temporal dependencies in the traffic network.

Benefiting from the ability of Transformers to capture long-range dependencies, some research has integrated GNNs with Transformers to capture the long-term spatial-temporal dependencies of traffic data [26,27]. Ref. [28] proposes a hierarchical framework that combines transformer networks and saptio-temporal graph convolutional networks to simultaneously capture the long-term temporal dependencies and short-term temporal and spatial dependencies within traffic data. Ref. [29] proposes a novel bidirectional spatial-temporal adaptive transformer model, which improves the accuracy and efficiency of urban traffic flow forecasting by dynamically adjusting computational loads and utilizing the reconstruction of past traffic conditions.

However, these methods overlook the dynamics of spatial dependence in traffic data. Merely considering the distance or spatial position of roads, or learning a static relationship from traffic data, fails to accurately capture the dynamic changes in road dependencies. To address this, we have innovatively proposed an adaptive graph convolutional module. This module adaptively constructs unique spatial relationships for traffic data across various temporal scales.

## 3. Traffic Flow Forecasting Problem Formulation

Traffic flow data often includes the observation data $X\in R^{N\times T\times D}=\left\{x_{1},x_{2},\cdots ,x_{T}\right\}$ and the corresponding road network structure *G*, where $x_{t}\in R^{N\times D}$ describes the traffic flow at the *t*-th period. Here, *N* denotes the number of roads, and *D* represents the dimension of features associated with each road at a given time. The road network structure reflects the topological connection relationship between roads and can be defined as G=(V,E,A), the node set $V=\{v_{1},v_{2},\cdots ,v_{N}\}$ of graph *G* represents the roads in the traffic network. Each node $v_{i}$ corresponds to a road in the traffic network, and the edges $e_{ij}\in E$ between nodes represent the connectivity between roads. The topological structure of an undirected graph can be expressed as an adjacency matrix $A\in R^{N\times N}$, which is determined by the edge set E. If $e_{ij}\in E$, then $a_{ij}=1$, otherwise $a_{ij}=0$. In this paper, given *T* step historical observation data, the traffic flow forecasting task aims at learning a function f(·) to forecast the future values ${\hat{X}}_{T+h}$ at time step T+h. The problem can be formulated as follows:(1)$${\hat{X}}_{T+h}=f\left(X,G,\theta \right),$$
where θ denotes all learnable parameters.

## 4. TEA-GCN Model Architecture and Methodology

In this section, we delve into the details of our proposed transformer-enhanced adaptive graph convolutional network(TEA-GCN), a novel approach designed to address the challenges of capturing dynamic spatial dependencies and long-term temporal dependencies in traffic flow forecasting. As illustrated in Figure 1, TEA-GCN is composed of local-global temporal attention layers to capture both long-term and short-term temporal dependence of the traffic data, as well as adaptive graph convolutional layers to learn adjacency matrices that best represent the spatial dependencies at different time levels. In the following, we will first introduce the local-global temporal attention layer in Section 4.1, followed by the adaptive graph convolutional layer in Section 4.2, then present the predictor and objective function in Section 4.3, and conclude with a complexity analysis in Section 4.4.

### 4.1. Local-Global Temporal Attention Layer

To effectively extract the temporal correlation of traffic data, we design a local-global temporal attention module. This module is designed to address the limitations of traditional CNN or RNN based recursive methods, which often struggle with capturing long-term dependencies due to issues like vanishing or exploding gradients. Our approach sidesteps these challenges by employing a non-recursive mechanism that extracts both short-term fluctuations and long-term trends.

Local Temporal Dependency Extraction: We utilize the traditional one-dimensional temporal convolution to mine short-term temporal dependencies within traffic data.Global Temporal Dependency Extraction: We employ the Transformer architecture to model long-range temporal dependencies within traffic data.Local-Global Temporal Information Fusion: Then, we design the temporal information fusion module to automatically capture the importance of different temporal patterns and promote the collaboration at different time levels.

#### 4.1.1. Local Temporal Dependency Extraction

During morning and evening peak hours, traffic flow is substantial, and the traffic congestion on one road at a certain moment may affect traffic speeds at later times, so it is necessary to consider the short-term temporal dependence of traffic flow in these situation.

In this paper, we utilize one-dimensional temporal convolution to extract local temporal dependencies within traffic flow data, capturing more sharper changes. Concurrently, to prevent the issue of gradient decay and ensure the adequate information transmission, we employ gated temporal convolution. This approach includes an update gate that regulates the flow of information, maintaining the integrity of data transmission throughout the model.

Given the input representation $Z^{\left(l-1\right)}$, the gated temporal convolution can be formulated as follow,
(2)$$\begin{array}{cc} T^{\left(l\right)} & =R^{\left(l\right)}⊙Z^{\left(l-1\right)}\\ \; & =\sigma_{1}\left(Z^{\left(l-1\right)}★\Theta_{1}+b_{1}\right)⊙\sigma_{2}\left(Z^{\left(l-1\right)}★\Theta_{2}+b_{2}\right), \end{array}$$
where $\Theta_{1}$, $\Theta_{2}$, $b_{1}$ and $b_{2}$ are learnable parameters. ★ denotes the convolution operation, ⊙ represents the Hadamard product operator. $\sigma_{1}\left(\cdot \right)$ is the non-linear activation function and we choose tanh in this paper. And $\sigma_{2}\left(\cdot \right)$ is the sigmoid(·) activation function. The sigmoid(·) function in the updated gate transforms the element value between 0 and 1 to control the proportion of information flowing into the next layer.

#### 4.1.2. Global Temporal Dependency Extraction

Compared to the sequential processing manner of traditional CNN or RNN based methods, the Transformer architecture employs a different approach by utilizing attention mechanism. This allows the Transformer to process information in a parallelized fashion, effectively recognizing dependencies between any two positions in the sequence. This mechanism enables the Transformer to capture long-term dependencies within the sequence data more effectively. In this paper, we utilize the Transformer to capture global temporal dependencies within traffic flow data.

The global temporal dependency extraction module features a transformer encoder-decoder network architecture, which is composed of three encoding blocks and three decoding blocks. Each building block primarily comprises two sub-layers: a multi-head self-attention mechanism and a fully-connected feed-forward network. We also insert the residual connection and the layer normalization behind each sub-layer to prevent degradation in the network.

Specifically, for the *l*-th layer of the encoder, the (l−1)-th layer representation $H^{\left(l-1\right)}$ is projected to the representation $H^{\left(l\right)}$ through the following equation,
(3)$$\begin{array}{cc} \; & {\mathrm{head}}_{i}=\mathrm{Att}\left(H^{\left(l-1\right)}W_{i}^{q},H^{\left(l-1\right)}W_{i}^{k},H^{\left(l-1\right)}W_{i}^{v}\right)\\ \; & \mathrm{MultiHead}\left(H^{\left(l-1\right)}\right)=\mathrm{Concat}\left({\mathrm{head}}_{1},\cdots ,{\mathrm{head}}_{m}\right)W^{O}\\ \; & {\mathrm{Res}}^{\left(l\right)}=\mathrm{Ln}\left(\mathrm{MultiHead}\left(H^{\left(l-1\right)}\right)+H^{\left(l-1\right)}\right)\\ \; & H^{\left(l\right)}=\max\left(0,{\mathrm{Res}}^{\left(l\right)}U_{1}^{\left(l\right)}+c_{1}\right)U_{2}^{\left(l\right)}+c_{2}\\ \; & H^{\left(l\right)}=\mathrm{Ln}\left(H^{\left(l\right)}+{\mathrm{Res}}^{\left(l\right)}\right), \end{array}$$
where $W_{i}^{q}$, $W_{i}^{k}$, $W_{i}^{v}$, $W^{O}$, $U_{1}^{\left(l\right)}$, $U_{2}^{\left(l\right)}$, $c_{1}$ and $c_{2}$ are the learnable parameters. Scaled dot-product attention function $\mathrm{Att}\left(Q,K,V\right)=\mathrm{softmax}\left(\frac{QK^{T}}{\sqrt{d_{k}}}\right)V$ is used to learn the correlation between the two time steps. Concat(·) is the concatenated function and Ln(·) denotes the layer normalization.

It should be noted that the input of the first encoder layer is slightly different, as it takes the raw sequence X as input. Given that traffic flow data is a typical time series, it is essential to incorporate the temporal position information of traffic flow data when modeling temporal dependencies. To fully leverage the sequence order, the transformer uses sine and cosine functions of different frequencies to add positional encodings to the input embeddings in the first layer of the encoder,
(4)$$\begin{array}{cc} {\mathrm{PE}}_{pos,2i} & =\sin(pos/{10,000}^{2i/d})\\ {\mathrm{PE}}_{pos,2i+1} & =\cos(pos/{10,000}^{2i/d}), \end{array}$$
where pos denotes the position and *i* is the dimension of the each element in the sequence data.

However, the above-mentioned position embedding method exhibits long-range periodicity, which makes it less effective at capturing the periodic patterns hidden in traffic flow data. To solve this problem, we propose a new temporal position embedding function with a given period,
(5)$${\mathrm{PE}}_{k}=\sin\left(2\pi k/period\right),$$
where *k* denotes the position of each element in the sequence, and period is the predefined period.

Subsequently, we incorporate the positional embedding into the corresponding data embedding, so that the proposed method can effectively encode the temporal position information within the traffic flow data. By this way, the input sequence $\tilde{X}$ of the first layer in the encoder part can be represented as the combination of the raw input X and the corresponding positional embedding PE,
(6)$$\tilde{X}=X+\mathrm{PE}.$$

The decoder in the global temporal dependency extraction module has the symmetrical structure with the encoder, which reconstructs the raw sequence from the embedding representation $H^{\left(\frac{L}{2}\right)}$. It is worth noting that the input to each layer in the decoder stack consists of the output $H^{\left(\frac{L}{2}\right)}$ from the last encoder layer and the output from the previous decoder layer. Therefore, the last layer of decoder can be formulated as,
(7)$$\begin{array}{cc} \; & {\mathrm{head}}_{i}=\mathrm{Att}\left(H^{\left(\frac{L}{2}\right)}W_{i}^{q},H^{\left(\frac{L}{2}\right)}W_{i}^{k},H^{\left(L-1\right)}W_{i}^{v}\right)\\ \; & \mathrm{MultiHead}\left(H^{\left(L-1\right)}\right)=\mathrm{Concat}\left({\mathrm{head}}_{1},\cdots ,{\mathrm{head}}_{m}\right)W^{O}\\ \; & {\mathrm{Res}}^{\left(L\right)}=\mathrm{Ln}\left(\mathrm{MultiHead}\left(H^{\left(L-1\right)}\right)+H^{\left(L-1\right)}\right)\\ \; & H^{\left(L\right)}=\max\left(0,{\mathrm{Res}}^{\left(L\right)}U_{1}^{\left(L\right)}+c_{1}\right)U_{2}^{\left(L\right)}+c_{2}\\ \; & \hat{X}=\mathrm{Ln}\left(H^{\left(L\right)}+{\mathrm{Res}}^{\left(L\right)}\right). \end{array}$$

Finally, we utilize the learned middle-layer embedding representation $H^{\left(\frac{L}{2}\right)}$ to forecast the traffic flow, and minimize the difference between the forecast traffic flow and its corresponding ground truth Y to train the global temporal dependency extraction module. Concurrently, to retain the characteristics of the raw sequence as much as possible, we also minimize the reconstruction error between the reconstructed sequence $\hat{X}$ and the raw sequence X. The corresponding loss function can be formulated as,
(8)$$\begin{array}{cc} L_{re} & =∥\hat{X}{-X∥}_{1}\\ L_{pre} & =∥\mathrm{Conv}\left(H^{\left(L\right)}\right){-Y∥}_{1}, \end{array}$$
where Conv(·) represents 1-D temporal convolution.

#### 4.1.3. Local-Global Temporal Information Fusion

In practice, the correlations between different time steps hidden in traffic flow data are very complex. The local temporal dependency extraction module uses the gated temporal convolution to capture the short-term temporal correlation of data, which is beneficial to analyze the short-term changing trend of traffic flow affected by emergencies. While the global temporal dependency extraction module exploits the attention mechanism to capture the long-term temporal dependence of data, which is conducive to analyze the periodic patterns in traffic flow data.

These two modules are specifically designed to analyze the changing trends of traffic flow under different temporal patterns. Therefore, to fully benefit from the advantages of these two modules and enhance the robustness of the proposed model, we creatively design a local-global information fusion module to respectively weight and fuse these two kinds of information layer-by-layer. In this way, the proposed model can fully explore the abundant and implicit temporal dependencies hidden in traffic flow data.

Specifically, for the *l*-th layer, we concatenate the representations $T^{\left(l\right)}$ learned from the local temporal dependency extraction module and the corresponding representation $H^{\left(l\right)}$ learned from the global temporal dependency extraction module through the following equation,
(9)$$Y^{\left(l\right)}=\mathrm{Conv}\left(\mathrm{Concat}\left(T^{\left(l\right)},H^{\left(l\right)}\right)\right).$$

We learn the weights to each element of $Y^{\left(l\right)}$ according to their corresponding importance, so that the fusion representation can focus on the information that is more critical to the current task and filter out irrelevant information. The fusion operation is formulated as,
(10)$$F^{\left(l\right)}=\mathrm{Att}\left(T^{\left(l\right)},H^{\left(l\right)},Y^{\left(l\right)}\right).$$

Then, we transfer the fusion representation $F^{\left(l\right)}$ to the corresponding adaptive graph convolutional layer to facilitate mining the spatial dependencies at different temporal levels.

### 4.2. Adaptive Graph Convolutional Module

The dependencies between roads change dynamically over time, and the explicit road network structure cannot accurately reflect the real-time dependencies between roads. Existing spatial-temporal graph convolution models learn a fixed adjacency matrix to represent the dependencies between roads, which fails to accurately capture the dynamic changes of the relationship between roads. To address this challenging problem, we creatively propose an adaptive graph convolutional module that automatically learns the dynamic spatial dependencies of traffic flow data layer-by-layer in an end-to-end manner.

Specifically, for the *l*-th adaptive graph convolutional layer, we formulate the adaptive graph convolution operation as,
(11)$$\begin{array}{cc} Z^{\left(l\right)} & =\sum_{k=0}^{K}P_{a}^{k}F^{\left(l\right)}W_{k1}^{\left(l\right)}+P_{b}^{k}F^{\left(l\right)}W_{k2}^{\left(l\right)}\\ \; & +A_{adp}^{k}F^{\left(l\right)}W_{k3}^{\left(l\right)}+{\left(A_{att}^{\left(l\right)}\right)}^{k}F^{\left(l\right)}W_{k4}^{\left(l\right)}, \end{array}$$
where $W_{k1}^{\left(l\right)}$, $W_{k2}^{\left(l\right)}$, $W_{k3}^{\left(l\right)}$ and $W_{k4}^{\left(l\right)}$ are the weight matrix of the *l*-th layer. The main difference between the proposed model and others lies in the topological structures, which is divided into four parts: $P_{a}$, $P_{b}$, $A_{adp}$ and $A_{att}$.

The matrices $P_{a}$ and $P_{b}$ represent the fixed physical connections between the roads, where $P_{a}=A/\mathrm{rowsum}\left(A\right)$ and $P_{b}=A^{T}/\mathrm{rowsum}\left(A^{T}\right)$.

The matrix $A_{adp}\in R^{N\times N}$ denotes the parameterized adjacency matrix. In contrast to the fixed physical structure, there are no constraints on the values within $A_{adp}$, and the elements in $A_{adp}$ can be optimized along with the network parameters during the training procedure. By this way, the proposed model can learn a graph structure that is tailored for the current traffic flow prediction task.

The matrix $A_{att}^{\left(l\right)}\in R^{N\times N}$ represents a data-dependent similarity graph that captures the interaction relationships between roads. To obtain the adaptive graph for the *l*-th layer, we calculate the inner product of the fused representation to confirm whether there is a correlation between two roads and how strong this correlation is,
(12)$$A_{att}^{\left(l\right)}=\mathrm{softmax}\left(\mathrm{Relu}\left(F^{\left(l\right)}\cdot {\left(F^{\left(l\right)}\right)}^{T}\right)\right),$$
where we utilize the softmax function to normalize the similarity matrix.

Different adjacency matrices serve distinct roles, and their combination can effectively capture the rich and implicit spatial dependencies within traffic flow data across various temporal levels.

### 4.3. Predictor and Objective Function

Finally, we obtain the optimal representation Z and send it to a predictor to generate the final traffic flow prediction results. In addition, to minimize the loss of information, we utilize both the mean absolute error and the mean square error to measure the difference between the prediction result and its corresponding ground truth values,
(13)$$\begin{array}{cc} L_{mae} & ={∥\mathrm{Conv}\left(Z\right)-Y∥}_{1}\\ L_{mse} & ={∥\mathrm{Conv}\left(Z\right)-Y∥}_{2}. \end{array}$$

By combining the reconstruction loss $L_{re}$, the prediction error $L_{pre}$, the mean absolute error $L_{mae}$, and the mean square error $L_{mse}$, we derive the total objective function for the proposed model as follows,
(14)$$\min L=\lambda_{1}L_{re}+\lambda_{2}L_{pre}+\lambda_{3}L_{mae}+\lambda_{4}L_{mse},$$
where hyper-parameters $\lambda_{1}$, $\lambda_{2}$, $\lambda_{3}$, and $\lambda_{4}$ balance the importance of different losses.

### 4.4. Complexity Analysis

For the input data features $X\in R^{N\times T\times C}$ and the fixed road network structure $A\in R^{N\times N}$, *N*, *T* and *C* respectively denote the number of sensors, the sequence length and the feature dimension. $C_{in}$ and $C_{out}$ respectively represent the number of input and output channels of the network. *K* denotes the kernel size of the temporal convolutional network. *D* is the mapping dimension of the transformer.

The major time-consuming burdens of the proposed method consists of the four main modules:For the Local Temporal Dependency Extraction module, the main computational complexity is $O(N\times \left(T-K+1\right)\times K\times C_{in}\times C_{out})$ due to the convolution operations.The Global Temporal Dependency Extraction module is mainly composed of the transformer encoder-decoder network, thus its complexity is $O(N\times T\times D^{2}+N\times D\times T^{2})$.For the Local-Global Temporal Information Fusion module, it mainly performs a convolution operation and an attention calculation, so its complexity is $O(N\times T\times C_{in}\times C_{out}+N\times D\times T^{2})$.The Adaptive graph convolutional module mainly conducts the matrix multiplication operation, thus its complexity is $O(N^{2}\times T\times C_{in}+N\times T\times C_{in}\times C_{out})$.
Overall, Since *K* is regarded as a constant, the total complexity of the proposed method is about $O(N^{2}\times T\times C_{in}+N\times T\times C_{in}\times C_{out}+N\times T\times D^{2})$.

## 5. Experimental Evaluation and Performance Analysis

In this section, we evaluate our proposed TEA-GCN model on two widely-used traffic datasets and compared it with 8 baseline methods. First, we introduce the experimental settings in Section 5.1, followed by the experimental results analysis in Section 5.2.

### 5.1. Experimental Settings

#### 5.1.1. Datasets

We evaluate the proposed method on three type of traffic datasets, including one road network traffic flow dataset PeMSD7(M) [30] and one metro passenger flow dataset Beijing Metro [31].

**PeMSD7(M)** (https://pems.dot.ca.gov/?dnode=Clearinghouse, accessed on 30 October 2024) collects the traffic speed data from 228 sensors deployed in District 7 of California, and the period is the weekdays from May to June in 2012 with a time interval of 5 min. Such dataset is a popular benchmark in the traffic forecasting tasks.**Beijing Metro** captures the passenger flow data from 325 stations of Beijing metro system during August 2015, with a time interval of 5 min.**PeMS-BAY** collects the driving speed data from 325 sensors located in the Bay Area, covering the period from 1 January 2017 to 31 May 2017.

#### 5.1.2. Compared Methods

Eight state-of-the-art traffic flow forecasting methods are chosen as the baseline,

**LVSR [32]** utilizes the support vector regression to predict the traffic flow.

**FNN [17]** is an auto-encoder network designed to learn compressed representations of the traffic data, which can then be used for forecasting.

**FC-LSTM [33]** exploits long short-term neural network (LSTM NN) to capture the long-term temporal dependencies of traffic data.

**STGCN [20]** models the traffic network as a graph, and uses the spatio-temporal graph convolutional networks to extract spatio-temporal correlations features from the traffic data.

**DCRNN [13]** combines diffusion convolution and gated recurrent units (GRU) to capture the spatial and temporal dependencies of traffic flow data, respectively.

**GWN [16]** captures implicit spatial dependencies hidden in traffic data by learning an adaptive graph, and uses the dilated casual convolution to capture temporal dependencies.

**STSGCN [14]** adopts a spatial-temporal synchronous graph convolution module to capture local spatial-temporal dependencies from traffic data.

**STFGNN [22]** combines the gated dilated CNN module and the spatial-temporal fusion graph module to simultaneously capture the local and global correlations of traffic data.

We choose three commonly-used evaluation metrics to measure the performance of our proposed method, i.e., Mean Absolute Error (MAE), Mean Absolute Percentage Error (MAPE) and Root Mean Square Error (RMSE). Since Beijing Subway is idle between 23:00 and 5:00, the Beijing Metro dataset exists a large number of 0. Thus, we cannot calculate its MAPE. It is worth noting that the lower the values are, the better the performance is.

#### 5.1.3. Parameters Settings

Following the work in [28], we partitioned the PeMSD7(M) dataset into a ratio of 7:1:2, and the Beijing Metro dataset into a ratio of 8:1:1, to form the respective training, validation, and testing sets.

For the comparison methods, we executed the original code obtained from the authors’ personal homepages and followed the parameter settings reported in the original papers to ensure optimal experimental results. Specifically, LSVR employs a linear kernel with a penalty term of 0.001. The FNN model consists of a three-layer fully connected network with dimensions of 12-128-64. The FC-LSTM model comprises a two-layer stacked LSTM network. The STGCN model consists of multiple ST-Conv blocks, each ST-Conv block featured three hidden layers with dimensions of 64-16-64. Both the size of graph convolutional kernels and temporal convolutional kernels are set to 3. The GWN model is composed of eight layers with a sequence of dilation factors 1, 2, 1, 2, 1, 2, 1, 2. The STSGCN model comprises four spatial-temporal synchronous graph convolutional layers (STSGCLs), with each STSGCL incorporating three graph convolutional operations, each using 64 filters. The STFGNN model includes three spatial-temporal fusion graph neural layers (STFGNLs), each consisting of eight independent spatial-temporal fusion graph neural modules (STFGNMs) and one gated convolution module with a dilation rate of 3. All models are optimized using the Adam optimizer with a learning rate of 0.001.

Our model has three spatial-temporal blocks and one predictor. The dimension of network layer is set as 1-32-32-32-64-128-1. This architecture was determined after a series of ablation studies where we varied the number of layers to balance model complexity and forecasting accuracy. $\lambda_{1}$, $\lambda_{2}$, $\lambda_{3}$, and $\lambda_{4}$ balance the corresponding loss items in the objective function, and we set values of these hyper-parameters as 0.001, 0.5, 1.0 and 0.1, respectively. We use the RMSProp optimizer to train our model, where the learning rate is set to 0.001. We set the batch size of PeMSD7(M) and Beijing Metro to 20 and 16, respectively. To alleviate the overfitting problem, we set the dropout to 0.3.

All the experiments are implemented in the environment of PyTorch 1.9.0 version, and carried out on a workstation with NVIDIA RTX 3090 GPU manufactured by NVIDIA Corporation, Santa Clara, CA, USA, AMD Ryzen 9 5900X 12-Core Processor manufactured by Advanced Micro Devices, Inc., Santa Clara, CA, USA, and 128G RAM manufactured by Kingston, CA, USA.

### 5.2. Experimental Results Analysis

We validate the proposed model on two datasets, and the results are shown in Table 1, Table 2 and Table 3. Obviously, the proposed model achieves the best prediction performance. The specific experiment results are analyzed as follows.

LSVR, FNN, and FC-LSTM aim to uncover the underlying patterns of traffic flow changes within historical data by modeling the temporal dependencies of traffic data. However, these temporal based methods often overlook the complex spatial structure inherent in traffic data. Obviously, the traffic speed on one road is not only determined by its own historical data, but also affected by the traffic speeds on adjacent roads. Therefore, it is difficult to accurately analyze the traffic flow trends by considering only the temporal dependencies within a single road segment. The experimental results show that the prediction performance of spatial relationship modeling methods (e.g., STGCN, DCRNN) is significantly better than that of purely temporal based methods, which fully proves the effectiveness of GNNs in capturing the complex spatial-temporal dependencies of traffic data.

The GNN-based traffic flow prediction methods integrate Graph Neural Networks (GNNs) with Convolutional Neural Networks (CNNs) or Recurrent Neural Networks (RNNs), aiming to capture both the temporal and spatial correlations within traffic data simultaneously. However, these methods overlook the dynamic of spatial dependence of traffic data over time; the traffic speed on one road at different times is affected by the correlation of different roads. The traffic data exhibit complex and implicit spatial correlation relationships. Focusing solely on the physical connections between roads or learning a static correlation relationship from the data fails to fully capture the dynamic characteristics of spatial dependence of traffic data, which leads to the sub-optimal performance of traffic flow prediction models.

We creatively propose a novel adaptive graph convolutional module to alternately capture the spatial and temporal dependencies of traffic data. This module learns a unique spatial structure for traffic data at various time levels in an end-to-end manner, effectively capturing the rich and implicit spatial dependencies inherent in traffic data. At the same time, our proposed local-global temporal attention module synchronously explores the long-term and short-term temporal dependencies hidden in the traffic data in a non-recursive manner, which can capture the temporal dependencies at different temporal levels for different tasks and effectively improve the robustness of the model.

## 6. Conclusions

In this paper, we propose a novel transformer-enhanced adaptive graph convolutional network for the traffic flow forecasting task, which contains two important modules. Specifically, an adaptive graph convolutional module is designed to capture the dynamic road spatial dependencies, and a local-global temporal attention module simultaneously captures the long-term and short-term temporal dependencies of traffic data. Our proposed model is able to alternately learn the temporal and spatial correlations layer-by-layer, which better reflects the spatio-temporal dependencies among roads and effectively improves the robustness of the proposed model. The traffic flow prediction results on two traffic datasets verify the superiority of the proposed model. In the future, we plan to focus on the adaptability of the model to various urban settings and integrates with additional data sources such as weather conditions, event calendars, or social media data to further enhance the predictive accuracy.

## Figures and Tables

**Figure 1 sensors-24-07086-f001:**
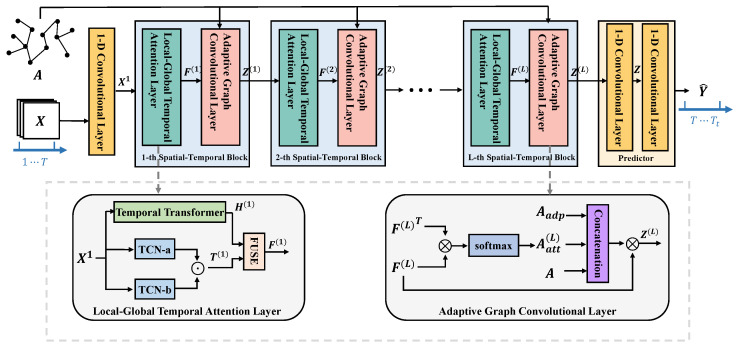
The brief illustration of the proposed model.

**Table 1 sensors-24-07086-t001:** Performance comparison on the PeMSD7(M) traffic flow dataset. We mark the best-performing results by bolded font.

Models	15 min			30 min			60 min		
	MAE	MAPE (%)	RMSE	MAE	MAPE (%)	RMSE	MAE	MAPE (%)	RMSE
LSVR	2.49	5.91	4.55	3.46	8.42	6.44	4.94	12.41	9.08
FNN	2.53	6.05	4.46	3.73	9.48	6.46	5.28	13.73	8.75
FC-LSTM	3.57	8.60	6.20	3.92	9.55	7.03	4.16	10.10	7.51
STGCN	2.25	5.26	4.04	3.05	7.33	5.70	4.04	9.77	7.55
DCRNN	2.37	5.54	4.21	3.31	8.06	5.96	4.01	9.99	7.19
GWN	2.82	6.80	4.80	3.90	8.93	6.94	4.89	12.90	9.01
STSGCN	2.59	6.19	4.91	3.34	8.18	6.59	4.62	11.71	8.75
STFGNN	2.47	5.86	4.54	3.23	8.10	6.27	4.21	10.35	8.07
TEA-GCN	**2.10**	**4.90**	**3.96**	**2.80**	**6.98**	**5.42**	**3.72**	**9.72**	**7.04**

**Table 2 sensors-24-07086-t002:** Performance comparison on the Beijing Metro flow dataset. We mark the best-performing results by bolded font.

Models	15 min		30 min		45 min	
	MAE	RMSE	MAE	RMSE	MAE	RMSE
LSVR	14.71	25.12	16.55	31.33	17.75	32.93
FNN	11.01	23.61	14.46	31.22	18.78	40.75
FC-LSTM	10.76	21.22	12.27	22.33	12.86	23.74
STGCN	7.83	16.81	9.56	17.92	10.16	20.29
DCRNN	8.37	19.13	9.46	23.38	11.63	25.87
GWN	11.91	24.64	14.24	29.24	16.14	36.64
STSGCN	10.65	20.71	12.24	24.03	16.22	33.23
STFGNN	9.13	17.47	9.06	18.50	11.72	22.39
TEA-GCN	**7.01**	**15.32**	**7.79**	**17.25**	**8.92**	**19.21**

**Table 3 sensors-24-07086-t003:** Performance comparison on the PeMS-BAY traffic flow dataset. We mark the best-performing results by bolded font.

Models	15 min			30 min			60 min		
	MAE	MAPE (%)	RMSE	MAE	MAPE (%)	RMSE	MAE	MAPE (%)	RMSE
LSVR	1.85	3.80	3.59	2.48	5.50	5.18	3.28	8.00	7.08
FNN	2.20	5.19	4.42	2.30	5.43	4.63	2.46	5.89	4.89
FC-LSTM	2.95	4.81	4.19	3.97	5.25	4.55	4.74	5.79	4.96
STGCN	1.39	3.00	2.92	1.84	4.22	4.12	2.42	5.58	5.33
DCRNN	1.38	2.90	2.95	1.74	3.90	3.97	2.07	4.92	4.74
GWN	1.30	2.69	2.74	**1.64**	**3.63**	**3.72**	**1.93**	**4.53**	**4.46**
STSGCN	1.57	4.34	4.42	1.98	4.64	4.51	2.53	6.13	5.97
STFGNN	1.47	3.14	3.04	1.91	4.32	4.28	2.44	6.07	5.54
TEA-GCN	**1.28**	**2.64**	**2.72**	1.68	3.80	3.73	2.03	4.91	4.60

## Data Availability

The original contributions presented in this study are included in the article material. Further inquiries can be directed to the corresponding author.

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
