# Peer review of "TEA-GCN: Transformer-Enhanced Adaptive Graph Convolutional Network for Traffic Flow Forecasting"

_sensors, 2024, doi:10.3390/s24217086_

Round 1
Reviewer 1 Report
Comments and Suggestions for Authors
1.The abstract need to be further revised and ensure it is accessible to a broader audience. Besides, the summary of related work (in Section 2) should be further extended. The titles of sections 3-5 could be more concise and specific.
2. It is not suggested to put a figure directly under the section title (e.g., Figure 1 on page 3). Moreover, it was not properly cited in the manuscript. Similarly, both Table 1 and Table 2 are suggested to move to Section 5.
3. The authors highlighted the dynamic characteristics of spatial dependence of traffic data in Section 5. However, these characteristics were not thoroughly discussed with regard to the model validations on two typical traffic datasets.
4. All parameters were defined directly in Section 5.1.3 without any explanation. Besides, a brief introduction of eight forecasting methods was provided in Section 5.1.2. Detailed information of the aforementioned model establishment needs to be complemented; otherwise, the comparison in Table 1 and 2 could be meaningless.
5. In addition to the commonly-used evaluation metrics (i.e., MAE, MAPE and RMSE), a detailed comparison of computational efficiency for all models would be useful.
6. Some suggestions on future research would be helpful for audience, which might be complemented in Section 5 and Section 6.
Comments on the Quality of English LanguageThe English language needs further editing.
Reviewer 2 Report
Comments and Suggestions for Authors
Please find my comments below:
1) The Related Works section is very limited. It should be expanded vastly, with references to the latest research in the field.
2) The paper lacks recent references in general. The newest of referenced works are from 2021. It is anticipated that the field has seen great improvements since then.
3) Section 3, introducing the problem statement, needs some rework and more clarifications regarding how N x D collectively define the number of roads, what a "node" means (i.e., V of G) in the given context, and why the road network was associated with an undirected graph.
4) To ensure transparency and reproducibility of the experiments, implementational details and training configurations for all the compared methods should be provided either in the paper or in a supplementary resource.
5) Section 5.1.3 mentions a specific dropout rate to prevent the overfitting problem. The problem should be analyzed in detail, and the choice of rate should be justified.
6) Results presented in Tables 1 and 2 need a thorough discussion in the corresponding sections. What do they mean in the physical world? How does the claimed improvement on top of existing methods benefit the industry or scientific community?
7) To ensure the proposed model's success, it is recommended that the experiments be repeated for a few other datasets.
Round 2
Reviewer 2 Report
Comments and Suggestions for Authors
I see that my previous concerns have largely been addressed.
My only request for this second round of revisions is that the negative results obtained from the previously requested additional tests involving the PEMS-BAY dataset and the Graph WaveNet (GWN) method be included in the paper. The impact and contribution of negative results have been underestimated by the scientific community for decades, which we now understand was a mistake.
Author Response
Comments 1: My only request for this second round of revisions is that the negative results obtained from the previously requested additional tests involving the PEMS-BAY dataset and the Graph WaveNet (GWN) method be included in the paper. The impact and contribution of negative results have been underestimated by the scientific community for decades, which we now understand was a mistake.
Response 1: Thank you for your suggestion. We have included the negative results from our additional tests involving the PEMS-BAY dataset and the Graph WaveNet (GWN) method in the revised manuscript.